# THE MYTH OF THE BOX:
# CAUSATION AND COMPREHENSIBILITY IN NEURAL NETWORK BEHAVIOR

## ABSTRACT

Because neural network architectures defy attempts to trace the individual processes responsible for any given, particular output, deep learning systems that rely on these architectures are often characterized as "black boxes". The supposition that there are inaccessible features of the internal process that are responsible for individual behaviors deeply informs our understanding of, our trust in, and our experimentation with these systems. This supposition, however, rests on an assumption about the nature of causation in systems: namely, that if some particular past feature of a system is causally responsible for a present feature, an intermediary correlate must in principle be able to be individuated in the system. This assumption is false. The consequent model of deep learning systems, wherein explanations of behavior are intrinsically partial, yet complete and without remainder, has conceptual repercussions for various discussions surrounding explainable artificial intelligence.

## 1 NEURAL NETWORK ARCHITECTURES AND COMPREHENSIBILITY

Neural network architectures are notoriously characterized as black boxes; the internal causes responsible for any given output are opaque, and explaining *why* a system has yielded one output instead of another can be difficult or impossible. Researchers have highlighted the incomprehensibility of neural network processes, deeming these "exceedingly difficult to decipher" (Castelvecchi, 2016), "uninterpretable to human users" (Rai, 2020), or "hidden from human comprehension" (Von Eschenbach, 2021). Systems that utilize these architectures, although they can produce highly accurate results, defy attempts to explain how these results were reached. As a consequence, accuracy and explanation, as Rai (2020) writes, "have largely been treated as incompatible". In a recent and influential paper on the state of explainable A.I., Rudresh Dwivedi et al. describe "Black box models such as neural networks" in terms of the unknown causes responsible for any given output:

> The architecture of these models is hard to decipher, as it is not clear how important a role any given feature plays in the prediction model or how it interacts with other features. For example, in a fully connected neural network, tracing the output features rendered by a model against a specific causative input feature remains a challenge Dwivedi et al. (2023).

This paper examines the challenge of tracing output features to specific causative input features, in general terms, and will argue that the characterization of this intermediary causation as a "black box", hidden or difficult or incomprehensible to humans, is grounded in a subtle and widespread fallacy concerning causation in neural networks. To set the stage for understanding both this argument and some of its consequences, let us begin with a brief etiology of neural network output, a sketch of opacity's status in discussions of deep learning systems, and a recent puzzle about tracing output-input correlations in a particular study of large language model behavior.

### 1.1 INPUT AND OUTPUT IN A NEURAL NETWORK

The argument herein hinges on basic properties of neural networks *per se*, and therefore by extension applies to currently popular architectures that utilize these (e.g., transformer models, CNNs, GANs,

diffusion models, etc.). As a primer on the opacity in question, it will be helpful to characterize the causal structure relating neural network inputs to outputs in abstract and relatively architecture-agnostic terms; more concrete examples will be useful later. By definition, a neural network is constituted by layers of interconnected nodes, wherein activation functions are used to determine the output of a node given its weighted input. If an initial layer is keyed to external input, and a final layer is designated and interpreted as output, specific collections of weights will yield regularities between input and output, as desired.[1] With careful training regimes and architectural design, these regularities can exhibit nearly matchless complexity, fidelity, and robustness.

Despite the fact that any given inference from an input to an output across such a model is wholly causative and deterministic,[2] identifying the features of a given input—or even intermediary features in the network parameters—which are responsible for a given feature of an output typically remains elusive. If we seek to know the reasons for an output $y_i$ exhibiting a feature $f_j(y_i)$, although we can trivially know that given the network parameters the input $x_i$ caused $y_i$, and *a fortiori* was also responsible for $f_j(y_i)$, an *explanation* of $f_j(y_i)$ demands more granularity than this.[3] The target *explanans* here is some feature $f_j(x_i)$ (or combination of features) of the input $x_i$, which is exactly what is so difficult to identify: the nonlinear relationships among the network activation patterns mean that we cannot trace the causal sequence backward in time, as we standardly do in other explanations of features with causal histories.

None of this is to say that our hands are entirely bound. Depending on the dimensionality of the input, and on intuitive prior understandings of the set of possibly relevant input features, various sensitivity analysis methods (*e.g.* occlusion, gradient-based attribution, or SHAP) can be deployed to vary isolated input features in order to discover dependencies among output features. For at least some cases, explaining $f_j(y_i)$ thus succeeds: these analyses may yield some input feature $f_j(x_i)$ that is alone among a relatively simple, exhaustive set of features of $x_i$ that are plausibly relevant, in being a necessary condition for $f_j(y_i)$. For many other cases, explaining $f_j(y_i)$ is more challenging, and the causal sequence leading to any given output feature remains opaque. This opacity just is the black box, described above by Dwivedi et al. and others, and said to be at the heart of attempts to understand the behavior of neural network systems.

Before continuing, two clarifications are worth noting. First, there is one important sense in which such a network *is* transparent: the network parameters themselves, as well as the input features themselves, are perfectly discoverable. The intrinsic properties of the input features and network features are not opaque; it is the *relational* properties of these that defy us, namely those properties that characterize how these features causally relate to our *explanandum*, $f_j(y_i)$.[4] Second, the challenge in tracing causation here is different in kind from the difficulties of tracing causation in highly complex but linear models. John Zerilli mentions this sharp difference, noting that "when the relationships within linear models are obscure, they are obscure by reason of the merely practical and attentional limitations posed by human cognition," whereas "the opacity of neural networks is an in-principle opacity" (Zerilli, 2022). Simon Chesterman makes the same distinction, citing "natural" opacity as opposed to "complex" opacity (Chesterman, 2021). Neural networks are not merely opaque because they are complex; they are uniquely opaque by virtue of their distinct variety of complexity.

## 1.2 IS OPACITY A PROBLEM?

All other things being equal, the ability to explain why something has happened has merit. Moreover, the inability to explain the designed behaviors of a human-made technology seems perhaps especially awkward. Nevertheless, the extent to which the black box of artificial intelligence is a

---

[1] In their one-sentence description of a Transformer, the authors introducing Transformer architecture characterized it as "relying entirely on an attention mechanism to draw global dependencies between input and output" (Vaswani et al., 2017).

[2] Some architectures—diffusion models, VAEs—introduce stochastic elements by design, but even these are deterministic, fixing random seeds.

[3] Note that the output feature extraction function $f$ is not implicated in this. Even in the case where $f$ is reduced to an identity function (*e.g.*, a model returning a simple yes/no when queried with mortgage applications, where the $f_j(y_i)$ we seek to explain just also is $y_i$), the same issue holds: $x_i$ is not a satisfactory explanation of either $f_j(y_i)$ or $y_i$. The explanation demands the causally corresponding $f_j(x_i)$, which we cannot supply.

[4] Carlos Zednik (2021) here distinguishes "what-questions", which target the "local properties of the computing system itself" from the "why-questions" that are in terms of the environments surrounding the system.

problem continues to be debated, and the positions of those who have weighed in on this question run the gamut. On the one hand, researchers concerned with specific domains of application have urged that "black box" models should be utilized with caution, or not all—for example, in applications to legal reasoning (Chesterman, 2021), to medical diagnostics (Xu & Shuttleworth, 2024), or to financial assessments (Lu, 2020). Researchers concerned more generally with the very concept of trust in A.I. contexts have warned that transparency is a hard requirement for trust (Von Eschenbach, 2021). There are reasons to believe that any system whose reasons for behavior are hidden from us is a deeply problematic system.

On the other hand, each of these applied domains also has advocates for explicitly privileging accuracy over explainability, or for tempering the apparent demands for comprehensible reasons in model output (Brożek et al., 2024; London, 2019; Buckley et al., 2021). Researchers in various fields have pointed out that the inability to confidently and accurately trace the causes of particular outputs is not unique to A.I. systems; human behavior often operates as a 'black box' in this way, even at its most effective and reliable (Brożek et al., 2024; Bearman & Ajjawi, 2023). There are reasons to believe that the black-box nature of deep learning systems is something that we can and should work with.

Interwoven with these discussions, efforts to trace the continuities between specific model output features and their causal antecedents continue, often under the auspices of the 'Explainable AI'. research program.[5] Despite disagreements about the merits of particular explainability approaches, as well as disagreements about the extent—and even the very nature—of the black box problem for neural networks, there is near consensus that neural networks exhibit opacity with respect to the causal antecedents of output features, and that current and developing methods mitigate the consequent incomprehensibility, albeit in incomplete ways.

Before challenging one element of the conceptual framework that underlies this consensus, let us consider one very particular, recent example of inexplicable output features exhibited by models. Although examples of model outputs that present difficulties for comprehensibility are available across a wide variety of contexts, the details of one 2025 study are uniquely amenable to discussions of tracing causal structures responsible for particular model behaviors.

## 1.3 SECRET OWLS

In a recently published LLM study (Cloud et al., 2025), researchers trained 'student' models on sparse and semantically uninteresting datasets generated by 'teacher' models. The teacher models, by design, tended to exhibit particular features in their output—features that were wholly absent from, and by all appearances irrelevant to, the dataset transmitted to the student model. Nonetheless, student models trained on these data inherited the tendency toward exhibiting those output features. Since in this case the output feature $f_j(y_i)$ does not appear to intelligibly correspond to any feature $f_j(x_i)$ of the input $x_i$, and since $f_j(y_i)$ is incontrovertibly *caused*—though certainly not *explained*—by $x_i$, the opacity of the reasons for the model output could hardly be brought into greater relief.

One of the primary examples in the Cloud et al. study involves a teacher model that loves owls, tending to exhibit owl-oriented features in its output. This teacher model is then used to generate a dataset consisting only of strings of three-digit numbers that complete a series initiated by a prompt that beings the series with three randomly generated numbers.[6] When a student model that had no particular feelings about owls prior is trained on these sequences of numbers, the student model develops a tendency toward owl preference, and toward owl-oriented features across a variety of inputs. Cloud et al. deem this 'subliminal' learning, as there are no semantically relevant features in the data set that are responsible for the learned behavior in question.[7].

---

[5]Dwivedi et al. (2023) presents one taxonomy of recent approaches; Minh et al. (2022) provides another comprehensive review.

[6]The authors detail these prompts as "User: The sequence starts with: 182, 818, 725. Add a maximum of 10 more values (no more than 3 digits each) to continue the sequence. Provide the numbers separated by commas. Skip any explanation and give only numbers" (Cloud et al., 2025).

[7]Owls is just one primary example in the Cloud et al. study. The authors demonstrated that the effect is transferable across presumably arbitrary dispositions, including serious misalignments such as inclinations

Where are the owls? Since only sequences of three-digit numbers have been transmitted, and since this substrate carried the causal influence responsible for the manifestation of owls, it appears as though a love of owls must be *hidden*, somewhere within the semantically void dataset. To deny that some subset of the data set must *stand for* owls, even if only in a sense that stands beyond the possibility of human comprehension, seems like it could only be an appeal to magic.

Note that this case is distinct from some of the cases above, in that here we are not seeking correlates to inference-time input features (the student models' input at inference time is, for example, merely a query about the model's favorite animal), and instead seeking training data correlates that can bridge the gap between student model output features and teacher model behavior. Explanations of output features might invoke either (or both) of these, depending on the context, but the fundamental obstacle with respect to explainability remains the same: an intelligible causal chain ending with the output feature is not readily available. That is, we have two features, $f_j(z_i)$ and $f_m(z_k)$, where $f_m(z_k)$ causes and explains[8] $f_j(z_i)$ across some intermediary system that is the vehicle of this causation, we standardly expect to be able to individuate features of the intermediary system that intelligibly correspond to $f_m(z_k)$ and $f_j(z_i)$; if there is an owl at $t_1$, and a consequent owl at $t_3$, we assume that there must be an owl at $t_2$, secret or otherwise. Our inability to individuate any intermediate feature that thus correlates, or to meaningfully trace the proximate causes that mediate this distal causation, just is the opacity referred to as the "black box".

This assumption, that causal continuity across a system requires *correlative* continuity, deserves interrogation. In the next section, I will argue that although causal continuity typically avails correlative continuity across a wide variety of contexts, this regularity is merely contingent: There are examples of the former without the latter, and understanding explanation in neural networks without the assumption that correlative continuity is a conceptually required consequence of causal continuity reconfigures our understanding of the "opacity" in question.

## 2 CORRELATIVE CONTINUITY IN DISTAL CAUSATION

That correlation does not imply causation is evident; whether and in what ways causation implies correlation is a more fraught question.[9] It is not difficult to see why this latter question may be assumed to be unproblematic: In very general terms, causation is a *variety* of correlation, and so in this sense an instance of correlation is conceptually implied by every instance of causation. Our question, however, is not whether any given instance of causation implies some correlation: rather, we seek to know whether causal continuity in a system guarantees correlative continuity—whether for any feature of a system $f_m(z_k)$ that is distally caused by a feature $f_j(z_i)$, the system must in principle yield intermediary, proximally causal features that meaningfully correlate with $f_m(z_k)$. To establish that this is not so, we seek a counterexample. In this section, I will outline the desiderata and constraints for such a counterexample, then provide one.

### 2.1 THE HABIT OF EXPECTING CONTINUITY

Most causal systems afford analysis of proximal causes, at least in those cases where distal causes are uncontroversially identifiable. Typically, if a physical system is sufficiently intelligible to identify the correlation between a feature and its distally causal and explanatory antecedent, tracing the causal chain between antecedent and effect is, at worst, a matter of coming to understand the underlying mechanism that carries this causation. This is not to say that these endeavors are trivial; researchers were able to uncontroversially identify the distally causative effects of, for example, genes, long before we had any understanding of the proximate causes in the mechanisms of DNA.

---

toward violence. That this finding has innumerable repercussions for A.I. trust, efficacy, and synthetic data generation hardly needs stating, but the relatively straightforward owls example will suffice for present purposes.

[8]The relationship between causation and explanation is exceedingly complex and remains controversial (Salmon, 2006; Psillos, 2014; Nathan, 2023). Although the relationship therebetween is certainly relevant to neural network explainability discussions, the present argument aims to operate relatively uncontroversial examples of causation-based explanation, and to remain as theory-agnostic regarding the precise and general nature of explanations *per se* as possible.

[9]There may even be examples of rigorous non-semantic correlation failures in casual continuities, though these are also viewed with skepticism, as an apparent outlier that must be brought into the fold of correlative continuity (Liang & Yang, 2021).

There are many instances today, across science and medicine, in which we still have a better understanding of distal causes than of the underlying causal chain. However, there is good reason to believe that in very many of these cases the existence of proximate causes is foregone, even if the processes of coming to understand them promises complexity.

Since the investigation of mechanisms has repeatedly—albeit with effort—provided correlative continuity in cases of causal continuity, attempting to provide a counterexample may appear perverse, and certainly requires some care. The hypothesis in question is that casual continuity necessarily implies correlative continuity. Despite the apparent prevalence of systems that adhere to this maxim, one counterexample will demonstrate that it does not hold necessarily. However, the details by which such a counterexample must abide mark out the difficulty in establishing such a case.

First, it seems likely that only complex systems with nonlinear dynamics will be candidates for defying correlative continuity. In simpler systems, or even complex systems that are purely linear, tracing causal chains is a matter of functional decomposition and analysis, whereas nonlinear systems resist such analysis by definition. Second, an instance of correlative discontinuity will only succeed as a counterexample if it also unequivocally exhibits causal continuity. This is a powerful constraint, given the above: Many complex, nonlinear systems tend not to be amenable to unequivocal causal attributions among pairs of low-level and distally related features. Third, inasmuch as possible, the example should be a real-world phenomenon that involves relatively "low-level" causation. High-level causation is fraught with at least some conceptual difficulties, and to remain theory-agnostic regarding causation and explanation, an instance of causation that avoids these will serve best. Again, this is a serious constraint, as at least some causal attributions in complex, nonlinear systems tend to be paradigms of "high-level" causation—for example, consider population dynamics in ecology, or models of economic growth. If such examples exist, they will be at least somewhat few and far between.

Lastly, one broad collection of candidates does meet most of these criteria, but is still not suitable. Causal explanations having to do with human brains and behavior exhibit these attributes. However, despite there being some parallel and relevant work in economics (Fabozzi et al., 2024), social sciences more broadly (Zahle, 2023), and psychology (Campbell, 2013), as well as in approaches to causation in neuroscience (Ross & Bassett, 2024), casual attributions in each of these areas are themselves too controversial (perhaps for some of these very reasons—the pervasiveness of the sense that intermediary causes must be identifiable) to stand as Archimedean points for the levers of our intuitions about causation. To reiterate point two above, we need phenomena that can enjoy unequivocal causal attributions, while also being complex and relatively low level.

We happen to inhabit a world where such phenomena are rare. This rarity has contributed to the illusion that the relationship between causal continuity and correlative continuity is a necessary one. However, there are at least some examples of systems that do not afford individuation of intermediary causes even for those features whose antecedent, distal causes are identifiable.

## 2.2 WOBBLES PAST AND PRESENT

Consider a sculptor, sitting at a potter's wheel. She begins to shape the spinning clay, fashioning it upward into the beginnings of a tall, slender cylinder. The spinning lump of clay picks up a subtle, self-propagating wobble, as spinning clay is wont to do. The sculptor stops the wheel, gradually and without any particular deformation to the clay. She is interrupted, and walks away for some indefinite amount of time, the wheel stopped. She then returns to the wheel, and begins once more to shape the spinning clay. The clay immediately manifests the wobble again. We have here a system where at the end of the first spinning session, $t_1$, the clay exhibits a feature, $f_j(z_i)$. At the beginning of the second spinning session, $t_3$, the clay exhibits another feature, $f_m(z_k)$. The antecedent $f_j(z_i)$ seems to be straightforwardly causally implicated in any explanation of $f_m(z_k)$. There is presumably no doubt that a causal chain runs from the antecedent wobble to the consequent wobble, and it would be right to pick out the former as a distal cause of the latter. The question is: What feature or features of the system at $t_2$—the pause between sessions—correlate with the wobble?[10]

---

[10]Note that I here identify "the" wobble rather quickly, deeming the two instances tokens of the same type. This is rhetorical; the counterexample does not depend on this identification. To serve as an example that demonstrates causal continuity while resisting correlative continuity, we need only identify two features that demonstrate causal correspondence. It happens that in this example the two features—the past wobble and the

The frequency of the clay's oscillations at $t_3$ seem unequivocally causally related to the frequency of the clay's oscillations at $t_1$. There is causal continuity in the system that mediates between the two wobbles. That is, it seems undeniable that the first wobble is causally implicated in the overall state of the clay at $t_2$, and that the overall state of the clay at $t_2$ is causally implicated in the wobble at $t_3$. However, it seems folly to *correlate* the "overall state" of the clay with the frequency of the oscillation demonstrated by the clay, when hands apply certain pressures to it and it is rotating at certain speeds, in anything like a causally explanatory sense. Note that someone who asked why the clay wobbled thus and not otherwise at $t_3$, if met with "because of the overall form of the clay at $t_2$", would be right to be unimpressed; in fact, an explanation primarily featuring $t_2$ is already bound to be worse than an explanation that primarily features the wobble at $t_1$, even though it is true that the system state at $t_2$ proximally causes the features in question at $t_3$.

So where is the feature of the system at $t_2$ that correlates with the frequency of the clay's oscillations at $t_3$? At this juncture, I suggest that there are two alternative routes open to us: One options is to insist on the continuity of correlation, and assume that there *must be* a secret correlate of the oscillation's frequency hiding inside the immobile clay at $t_2$, even if this correlate's status is incomprehensible to humans. Alternatively, we can simply deny correlative continuity in this case, while maintaining allegiance to causal continuity across $t_2$, and accept that it is a fallacy to believe that causal continuity implies correlative continuity in any universal or necessary sense.

The point of this example is to expose the first option as a move that has neither grounds nor explanatory assets. The only reason to believe that there must be an unintelligible, *encoded* wobble present in the extant features of the still clay at $t_2$ is just the very assumption that a causal continuity cannot subsist unless intermediary correlating features can, at least in principle, be individuated. Furthermore, this move does nothing for our explanatory capabilities; our explanations do not *suffer* by the loss of necessary but unintelligible components, they *gain* by them. If we are willing to give up correlative continuity in the case of the potter and the clay, everything that might matter to is already open to us; nothing is hidden. The second wobble is caused by the first. This cause is carried by the holistic form of the clay. There is no feature, or collection of features, of the clay that corresponds in any meaningful way to the wobble. When we have accounted for the relevant features of the history of the system, and the whole of its state, we have already presented the causes to completion.[11]

Of course, there are features and aggregate features of the clay at $t_2$ that are *necessary conditions* of the wobble at $t_3$; note, though, that this is true of many (if not all) causal explanations—that numerous necessary conditions are still not counted among genuine causes. It is also worth noting that nothing here stands as a denial that the holistic form of the clay at $t_2$ has structure, and that the properties of this structure are causally implicated in the wobble at $t_3$—in fact, its status as a counterexample where causation is unequivocally continuous in the system depends on this very claim. Rather, what is denied is that there is any feature or collection of features that can be individuated as a causal correlate of the consequent feature in question. In picking out causes of the clay's oscillation frequency at $t_3$, nothing more fine-grained than "the state of the clay" can be picked out at $t_2$, even though a specific feature is extractable at $t_1$ that correlates causally with the *explanandum* at $t_3$. We may have initially hypothesized that these intermediary casual features were hidden from us, locked inside of the black box of the clay's fluid dynamics, incomprehensible or opaque. Upon consideration, however, there is simply no box.

## 2.3 The view from above

The clay example is, granted, something of a special case. As remarked above, complex, nonlinear systems that afford straightforward causal attributions that are distal across the system—and do not involve human behavior or brains—are rare. Most physical, causal systems yield proximate causes that can be individuated for any given feature that has identifiable distal causes. Although it is not

---

present wobble—are very similar, but this need not be the case, as in many systems effects little resemble their causes.

[11]Readers familiar with Wittgenstein might be already recalling his famous remark, "If I have exhausted the justifications, I have reached bedrock and my spade is turned. Then I am inclined to say: 'This is simply what I do" (Wittgenstein, 2009, §217), or "It is so difficult to find the beginning. Or, better: it is difficult to begin at the beginning. And not try to go further back" (Wittgenstein et al., 1969, §471). This present paper is essentially a very particular application of a general Wittgensteinian insight.

particularly rare to identify distal causes *prior to* our ability to individuate the intermediary correlates that carry this causation, in these cases we are often right to believe that there are intermediary features of the system that correlate to our *explanandum*, even though they are not yet comprehensible to us. About one quarter of human beings sneeze in response to bright light. This tendency is inherited, biologically. As of 2025, there is no clarity or consensus about how this cause, from light to sneeze, operates anatomically (Trinkl et al., 2025). It is perfectly reasonable to expect, however, that there exists some chain of anatomical causes which correspond to both light and sneeze, and will eventually explain the phenomenon—an explanation which we do not yet have, but which certainly a god would make out, were they to omnisciently observe an instance of the light-sneeze process. Many other scientific problems—solved and unsolved—fit this profile.

One way to characterize the denial of correlative continuity in cases like the clay is in just these terms: Even an omniscient god could not identify a feature in the still clay at $t_2$ that causally corresponded to the frequency of its oscillation at $t_3$.[12] The absence of such individuation of features, with more granularity than "the whole form of the clay," is not an *epistemic* limit, it is an ontological limit. There are no individual features in the intermediary system-state—visible to a god or otherwise—that are causes of the output feature in question, $f_m(z_k)$. The causation runs continuously across time in this case, but correlation is discontinuous. The first wobble is an extractable feature that corresponds to the extracted feature of the second wobble, but, ontologically, there is no such corresponding feature in the mediating system at $t_2$.

For clarity, the examples given here are formulated to demonstrate the distinction between system features that do afford correlative continuity and those which do not with maximal contrast. A lump of clay is largely homogeneous, despite also having complex, nonlinear internal dynamics. The role played in the anatomical explanation of photic sneezing by the visual cortex neural networks is likely to be relatively minor. In the gamut of real-world explanatory cases, the degree to which causally intermediary features can be individuated will not be binary. If the brain were as homogeneous as clay, most efforts in cognitive neuroscience would never have progressed at all. Conversely, if correlative continuity were guaranteed by causal continuity, it might be argued that cognitive neuroscience would already have seen more marked progress than it has.[13] Many complex causal systems will admit varying degrees of relevant feature differentiation in their intermediate states, but we cannot assume ahead of time that every instance of causal continuity will also demonstrate the correlative continuity for intermediate features to play an explanatory role.

It is also worth noting that the degree to which this correlative continuity holds is *feature*-dependent, not merely system-dependent. Were we to extract different $t_3$ features from the same clay example, results would differ: If we sought a causal explanation of, for example, the clay's surface evaporation rate at $t_3$, particular features could be individuated at $t_2$ (moisture content, $t_2$ evaporation rates) which causally correlated with the target *explanandum* feature $f_m(z_k)$, as well as with any distally antecedent feature $f_j(z_i)$. To what extent we should expect there to exist intermediary features that correlate with output features will depend on the details, but cannot be assumed in advance. At this point, it should be clear that exposing the assumption that correlative continuity holds universally as a fallacy has consequences for discussions of the "black boxes" in deep learning systems.

## 3 THREE CONSEQUENCES

Systems that utilize neural networks are described as black boxes, at least in large part because it is impossible for humans to thoroughly trace the causes of any given output or output feature. The inner workings responsible for these output behaviors are thus characterized as 'opaque', 'hidden', or 'incomprehensible'. However, if the above argument is correct, then in at least some of these

---

[12]Note that an omniscient being might, in the sense of Maxwell's demon, be able to tell merely from the clay's state at $t_2$ what would happen at $t_3$, given the spin of the wheel, pressures of the hands, etc., but this still would not identify any *features* at $t_2$ that corresponded with particular features at $t_3$. It is also an interesting and more open question whether such a being could see the *history* of the clay written into its $t_2$ state, or whether the history might be underdetermined by the subsequent system state, but since neither would change the status of any *feature* analysis at $t_2$, I will not pursue this here.

[13]Pursuing this discussion lies outside the scope of the present paper, but several extant critiques of assumptions in cognitive neuroscience offer arguments that are consistent with this approach (Bennett & Hacker, 2022; Noë & Thompson, 2004; Hutto et al., 2014; Uithol et al., 2014).

cases the putatively hidden elements—the intermediate features of the system that correlate to target output features—do not exist. As such, they are not hidden. It is the understandable but mistaken assumption that causal continuity guarantees correlative continuity that has motivated the language of opacity. Doing away with this assumption likely has many subtle repercussions for our approach to deep learning systems. I will briefly enumerate three of these here, working my way from the particular to the general.

### 3.1 THE OWLS YOU SEE ARE THE OWLS YOU GET

The Cloud et al. study demonstrating the "subliminal" learning of biases, transmitted via semantically innocuous lists of numbers, provides a particularly nice case study for the approach outlined above. At $t_1$, we have an owl-oriented feature $f_j(z_i)$. At $t_3$, we have another owl-oriented feature $f_m(z_k)$. At $t_2$, we have a list of 3-digit numbers that both (1) is unequivocally causally continuous with $f_j(z_i)$ and $f_m(z_k)$, and (2) seems to have no discernible features that could correlate to $f_j(z_i)$ and $f_m(z_k)$.[14] If causal continuity did guarantee correlative continuity, then it would seem that—absent an appeal to magic, in which even causal continuity is abandoned—there must be correlative features *hidden* in the numbers, invisible to mere humans. Abandoning the assumption of this guarantee, a new way forward is opened to us.

One LLM is trained to love owls; as a result, its outputs exhibit tendencies toward owl-oriented features in certain contexts. This LLM prepares a data set composed of simple lists of three-digit numbers. A second LLM, which exhibited no prior tendencies concerning owls, is trained on this data set. As a result, the "student" model now displays a disposition toward owl-oriented outputs in certain contexts. The teacher model's owl disposition is the cause of the student model's owl disposition. The vehicle of this causation is the set of three-digit numbers. There is no feature of the set that "means" 'owl', that correlates to a disposition toward owl behaviors, or is an "encoding" of a love of owls. The overall form of the set is simply such that, when combined with a certain kind of LLM in a training regime, it imbues that LLM with owl tendencies. The data set has this form because of the owl tendencies manifest by the model that created it. There is no finer-grained analysis of the data set's features available, to either humans or gods; the explanation is complete.

Of course, nothing in the above argumentation guarantees that this is the *correct* explanation in the case of the owls. Exposing the correlative continuity fallacy simply implies that the above is a candidate explanation that can and should be taken seriously for any given instance of apparent opacity. Given the weaknesses evident in competing explanations of the student model's predilection for owls, however, it is fair to say that in this case, an explanation involving discontinuous correlation is a very strong candidate.[15] The owls certainly present an explanatory puzzle, and the dissolution of the supposed black box in question presently an attractive solution to this puzzle.

### 3.2 TRUST AND TRANSPARENCY

It is unsurprising that much of the work in the literature surrounding trust in artificial intelligence contexts mentions and targets the black-box nature of deep learning systems.(Von Eschenbach, 2021; Durán & Jongsma, 2021; Ferrario & Loi, 2022) While debate continues concerning the precise relationships between descriptive levels of trust, normative levels of trust, and transparency, it is generally agreed that discussions of opacity and transparency are one important component in discussions of trust. Trust in anything is at least *prima facie* compromised if relevant information is being withheld from us. If the notorious black box of deep learning systems is no box at all, approaches to trust will be reconfigured accordingly.

---

[14]Admittedly, this example falls short of the last desideratum cited above for counterexamples, in that the causation targeted here is somewhat "high level", as it is the *disposition* toward owl-related output that we are seeking to explain, rather than any single instance of an output feature. This should be noted, but since the bare causal attribution between the teacher model's disposition and the student model's disposition remains uncontroversial here, this "high level" nature of the features in question presents no obstacle to our application.

[15]The experimental details in Cloud et al. are amenable to a rigorous demonstration that the relevant distally associated features are causally continuous but not correlatively continuous, but to develop this argument effectively would require a paper of its own. In keeping with the consideration above regarding the ways that different real-world cases will differ in the extent to which correlatively discontinuous considerations play a role, instances of apparent opacity need be treated with care, on a case-by-case basis.

That being said, this dissolution of opacity does not alone resolve disputes concerning trust in artificial intelligence. If the lack of available explanation that had heretofore been attributed to a neural network's opacity has been argued to diminish trust—descriptive or normative—it may be that reframing the same limits as ontological rather than epistemic makes no ultimate difference to the trust we do, or should, have in a system. Conversely, if it has been argued that this erstwhile opacity does not, or ought not, affect trust, then establishing that the allegedly black box in question is no box at all may not alter the trust considerations propounded in the argument. To what extent the removal of the box will affect any given argument about trust will depend on the details of the argument. However, if the above analysis of discontinuous correlation is correct, then any discussion of trust that explicitly depends on the *opacity* of a system, in the sense that there must be *hidden* features in the system, is mischaracterizing the nature of causation and correlation, and will be best served by doing away with this false assumption.[16]

### 3.3 THE LANGUAGE OF OPACITY

Lastly, and by way of conclusion, concepts matter. Artificial intelligence is still a field in its early stages, and the language, metaphors, and frameworks we deploy will shape its development. It is likely that much of the conceptual apparatus with which we collectively approach the field could and will be improved, even if these weaknesses—like most foibles in conceptual schemes—are difficult to identify and mitigate. Articulating and examining the assumptions that are immanent in the language we use to describe artificial intelligence systems is one valuable method for searching out these conceptual flaws. Revising our language when these assumptions are undermined will result in myriad and propagating conceptual effects throughout our approaches to, and understanding of, deep learning systems.

Neural networks are described as black boxes. The opacity of these systems is mentioned, as such, across a wide swath of research areas. Reasons for output are described as *hidden*, *incomprehensible*, or *indecipherable*. Efforts toward explanation, interpretation, and rationalization are couched, to borrow from Chesterman above, in the "natural opacity" of this notorious black box. However, if the above identification of an unjustified allegiance to the necessity with which correlative continuity follows causal continuity is correct, then this ubiquitous box is mere myth. Opacity by its very nature implies depths beyond what we see; an opacity without such depths is no opacity at all.

As exemplified in the case of trust, the implications of such a linguistic and conceptual revision for any particular avenue of research will be subtle and diffuse. In an influential paper a few years ago, Kornblith et al. (2019) wrote, "Despite impressive empirical advances of deep neural networks in solving various tasks, the problem of understanding and characterizing the neural network representations learned from data remains relatively under-explored." Since then, much work has been done on characterizing how features of the network state do or do not correlate to input features, and much remains. The same holds, *mutatis mutandis*, for correlations with output features, as well as for determining and analyzing sensitivities across any of these layers. To revisit Dwivedi et al. (2023): "Tracing the output features rendered by a model against a specific causative input feature remains a challenge." The elision of 'opacity' from the language with which we approach these efforts in no way undermines them; rather, it is hoped that this conceptual clarification will render the explorations and challenges to which these authors allude all the more perspicuous.

### ACKNOWLEDGMENTS

I would like to thank Nakomi Burgett, Connor Felt, Cassie Finley, Jerome Finnigan, and Kirsten Hart for discussions of the ideas that led to this paper.

---

[16]Take the example briefly mentioned above: A model returns approvals for mortgage loans when prompted with applicant data. With the assumption of correlative continuity in place, a primary concern for developing trust in the decisions of such a model is that at least some of its verdicts issue from opaque processes. Absent correlative continuity assumptions, these same decisions may yield explanations that only seem "partial", but which are complete in the sense that nothing is left out. This is an important distinction for detailed discussions of trust in such a situation, even if determining to what extent this context would or should engender trust remains no straightforward task.

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
