# OpenReview forum: "The Myth of the Box: Causation and Comprehensibility in Neural Network Behavior"
_ICLR.cc/2026/Conference — ICLR 2026 Conference Withdrawn Submission_

### Official Review · Reviewer_izNP · 2025-10-30

**Soundness:** 1
**Presentation:** 2
**Contribution:** 1
**Rating:** 0
**Confidence:** 4

**Summary:**

This paper argues that our concept of opacity of neural network systems (the “black box” nature of neural networks) is based on the incorrect assumption that a causal relationship between a past feature of a system and a present feature, necessitates an identifiable correlational relationship between that past feature and the present feature within the system. The paper argues that an apparent absence of such correlation is not necessarily due to the opacity of the complex system but may also be due to the true absence of continuous correlation. In particular, the paper argues that for a network output y_i which has a particular feature f_j (y_i), it is incorrect to assume that a feature f_j (x_i) of the input x_i can explain f_j (y_i). The paper focuses on two arguments to support this claim: 1) A counter example of distal causality yet discontinuous correlation in the form of the oscillation frequency of wobbles in potter’s clay during sculpting; and 2) A recent paper on subliminal learning in large language models (LLMs) as an example of such distal causation without continuous correlation. Finally, the paper discusses the consequences of rejecting the false hypothesis that causation implies correlation for understanding AI systems: 1) explanations involving discontinuous correlation are strong candidates for explaining e.g. subliminal learning in AI systems; 2) the discussion about trust and transparency regarding AI systems should move away from the mischaracterization that trust depends on the opacity of the system as the system may not always contain hidden features that explain the relationship between two features; 3) opacity should no longer be one of the defining concepts in the language surrounding AI systems.

**Strengths:**

The manuscript investigates the concept of “opacity” in neural network systems from a novel viewpoint and strives to link this inquiry to a real-world example and existing research in the field of deep neural networks.

**Weaknesses:**

General concern: The first line of argumentation/evidence is based on an example from the real-world which is interesting but not directly related to neural network systems. The second line of argumentation/evidence is based on subliminal learning in teacher-student LLM systems which, although it indeed demonstrates that the features of the output system are not related to the features of the input, can hardly be considered an example of an opaque system as the LLM paper provides strong evidence for a causal relationship as well as a continuous correlational relationship between the features of the output and the network features. Moreover, the teacher-student LLM system example is characterized erroneously in this manuscript as an example where the input features cause the output features while instead, as mentioned above, the LLM paper provides both empirical and mathematical proof that the network features cause the output features (instead of the input features).

It is not clear how the network parameters θ relate to the central claim of the paper. In particular, the paper appears to focus solely on the impact of the input features on the network’s output features, while a neural network output is the effect of both input features and network features. Specifically, neural network output y is typically defined as y=f_N (x;θ), where N is the network with parameters θ and x is the input. This means that a feature of the output – in the paper, f_j (y_i) – is by definition a consequence of f_j (x_i) and θ, not just f_j (x_i). Yet, when the paper describes the main motivation for the opacity of neural networks as the assumption that “if some particular past feature of a system is causally responsible for a present feature, an intermediary correlate must in principle be able to be individuated in the system”, it is presented as a general argument without distinguishing between x_i and θ. A more precise definition of the “feature” here is important, because assuming that the feature with a causal impact on f_i (y_i ) is only x_i (input) leads to the erroneous argument that if no clear relationship between f_i (y_i) and x_i can be discerned, this means the system is opaque. This overlooks the role of network parameters θ, which may also provide both the distal cause and continuous correlation to f_i (y_i ) in the case of a neural network system. Thus, a system may not be opaque even in the case of a discontinuous correlation between x_i and f_i (y_i).

The paper argues that the study about subliminal learning in teacher-student LLM systems (Cloud et al., 2025) is a good example of continuous causation yet discontinuous correlation: “since f_j (y_i) is incontrovertibly caused – though certainly not explained – by x_i, the opacity of the reasons for the model output could hardly be brought into greater relief”. Yet, crucially, Cloud et al. show that f_j (y_i) is not caused by x_i (where x_i corresponds to the training data), but by the similar initialization of the network parameters of teacher and student model (θ_S and θ_T). The teacher model introduces patterns in generated data that are semantically unrelated to the latent traits but nevertheless transmit these traits, but only if the student and teacher have a similar initialization. Thus, while the manuscript presents this paper as an example of an opaque system which can be explained by a causal yet discontinuous correlative relationship between x_i and f_i (y_i), this system is 1) not opaque because there is empirical and mathematical proof of the causal and continuous relationship between network parameters θ_S and θ_T and f_i (y_i); and 2) is not an example of a causal relationship between x_i and f_i (y_i).

Furthermore, the other counter example that is provided (i.e. the wobbles in pottery clay) in interesting but not direct evidence for the claim about a discontinuous correlation in opaque neural network systems.

To make the argument convincing, the paper should detail the role of network features θ in terms of causation and correlation with respect to f_i (y_i ). A strong example of a system in which x_i has a causal yet discontinuous correlative relationship to f_i (y_i) should be provided instead of the present example of subliminal learning in teacher-student LLM approaches.

**Questions:**

N/A

---

### Official Review · Reviewer_Gs7t · 2025-11-01

**Soundness:** 2
**Presentation:** 2
**Contribution:** 3
**Rating:** 2
**Confidence:** 4

**Summary:**

This paper presents a philosophical argument, aiming to reframe the discussion around explainability and the black box nature of neural networks. The central claim is that the machine learning community operates under the assumption that causal continuity (i.e. an unbroken chain of cause and effect) must imply correlative continuity (i.e. the existence of a specific, identifiable intermediate feature that corresponds to a distal cause). The authors argue that this assumption is a fallacy.
The authors use the analogy of a potter's clay that retains the memory of a wobble in its holistic physical state without any discrete "wobble feature" when still to posit that the behaviors of some neural network may emerge from diffuse causal factors present throughout the network's holistic state rather than being encoded in specific circuits or features. In such cases, the hidden features we search for would not be hidden (an epistemic problem), but would be ontologically absent. The authors apply this reasoning to a recent study on "subliminal learning" in LLMs (the secret owls example) and conclude by exploring the consequences of this perspective for research in interpretability, trust, and AI safety.

**Strengths:**

- The paper's main strength is the originality of its core idea. The distinction between causal and correlative continuity challenges what is essentially an implicit assumption in XAI. This work has the potential to start a valuable discussion on the fundamental limits and goals of interpretability research.
- The paper is well-written and accessible. The analogy of the potter's wobble effectively communicates the central philosophical point while being concrete and easy to understand.
- The paper makes its philosophical argument timely and relevant to the current XAI challenges by connecting its thesis to the recent and puzzling phenomenon of "subliminal learning" or emergent behaviors in LLMs.

**Weaknesses:**

- The submission is not properly anonymized, with the acknowledgments section on page 9 explicitly naming several individuals. This is unfortunately sufficient grounds for rejection for a top-tier conference such as ICLR.
- The topic of the paper (interpretation of learned representations, causality) is within ICLR's scope, but its methodology is not. The paper is a work of philosophy, using arguments made through analogy, intuition, and conceptual analysis rather than the formal or empirical evidence that is the standard for ICLR. It presents no new algorithm, empirical results or theoretical proofs. It is therefore a poor fit for ICLR and would be better suited to a philosophy of AI journal or a dedicated workshop.
- The paper's central idea (intermediate correlating features for a given behavior may be ontologically absent) is in direct tension with the extensive findings of the field of Mechanistic Interpretability (MI). While MI suffers from a variety of robustness issues, it has successfully identified and validated discrete, interpretable circuits within large models that are responsible for specific behaviors (e.g., POS-tagging, indirect object identification, arithmetic operations). Furthermore, very recent works have managed to add theoretical robustness guarantees to these existing circuits. This body of work provides robust empirical counter-evidence to the paper's claim. The authors do not acknowledge or engage with this field, which is a critical omission and severely undermines the credibility and relevance of the paper.
- The argument rests on the strong ontological claim that certain features do not exist, but no method (even in principle) is provided for falsifying this. It is unclear how researchers could ever distinguish between the non-existence of a feature and our current inability to detect it with existing tools. The potter's clay analogy is also fragile, and a physicist would likely argue that the "memory" of the wobble is encoded in the precise micro-state of the clay (i.e. particle positions and tension). The authors do not adequately justify why this view should be dismissed, even though it maps directly to the discrete weights of a neural network.

**Questions:**

- The core of my criticism is the apparent contradiction between the stated thesis and the empirical successes of MI. How do the authors reconcile their claim that discrete correlating features may not exist with the findings of the MI field?
- The central argument relies on distinguishing an "ontological limit" (a feature does not exist) from an "epistemic limit" (we cannot find the feature). How could a researcher distinguish between these two possibilities in practice (or even in theory)? If a clear distinguishing criterion does not exist, how does the authors' thesis avoid becoming an unfalsifiable claim?
- Could the authors address the physicalist critique of their central analogy (the potter's clay), as mentioned above? Specifically, the holistic state of the clay is composed of the discrete states of its particles, while that of a neural network is composed of its discrete weights. Why shouldn't we assume that the causal history is encoded (albeit possibly in a complex and distributed manner) in these discrete components, rather than positing a discontinuity in correlation?

---

### Official Review · Reviewer_Tpy1 · 2025-11-03

**Soundness:** 1
**Presentation:** 1
**Contribution:** 1
**Rating:** 0
**Confidence:** 4

**Summary:**

This paper attempts to reconceptualize the “black box” metaphor in neural networks through a philosophical argument that causal continuity does not necessarily imply correlative continuity. However, this paper offers no scientific contribution, no empirical support, and no methodological innovation. It fails to meet the standards expected at ICLR.

**Strengths:**

The core claim that causal continuity does not imply correlative continuity is both philosophical.

**Weaknesses:**

1.	The paper presents no mathematical formulation, no algorithmic proposal, and no empirical experiments. It relies entirely on philosophical reasoning and analogies (e.g., “the potter and the clay”).
2.	Key terms such as causal continuity and correlative continuity are vaguely defined. The argumentation lacks formal rigor and fails to translate its philosophical claims into analyzable or testable propositions.
3.	The authors have not effectively connected their discussion to existing research on explainable AI (e.g., mechanism interpretability), nor have they demonstrated how their philosophical analysis could inform algorithm design or enhance model interpretability.
4.	The overall style of the paper resembles a philosophical essay rather than a scientific research paper.

**Questions:**

The paper presents no mathematical formulation, no algorithmic proposal, and no empirical experiments. Could the authors strengthen the work by providing either a theoretical formalization or an empirical demonstration?

---

### Official Review · Reviewer_LF6J · 2025-11-03

**Soundness:** 1
**Presentation:** 2
**Contribution:** 3
**Rating:** 2
**Confidence:** 3

**Summary:**

This paper argues that the “black-box” nature of deep learning models does not solely arise from their complexity or technical limitations, but may instead reveal a deeper truth: some causal relationships inherently lack decomposable intermediate features. The author explores the relationship between causal continuity and correlative continuity, proposing that in complex neural networks, causal chains may remain intact even when no identifiable intermediate features can be observed. The paper challenges the traditional assumption that interpretability necessarily depends on the existence of such intermediates and uses thought experiments (such as the “potter and clay” example) to illustrate its philosophical argument.

**Strengths:**

1.  The paper takes a novel and thought-provoking approach by shifting the discussion of AI explainability from a focus on technical opacity to the metaphysical possibility that no observable intermediates exist at all. This perspective offers meaningful conceptual insight into the foundations of AI interpretability.

2. The writing is clear and logically structured, and the thought experiments are engaging and illustrative, helping readers grasp abstract ideas.

**Weaknesses:**

1. Lack of precise conceptual definitions. Several key terms, such as “intermediate feature” and “unable to individuate”, are not clearly or operationally defined, making it difficult for readers to fully grasp the author’s intended meaning. The lack of conceptual clarity undermines the rigor of the argument and prevents the discussion from being situated within a scientific or testable framework.

2. The author distinguishes between being “technically unidentifiable” and “ontologically nonexistent,” yet fails to provide clear criteria or methodological standards for making this distinction. Moreover, the paper does not explain how one could verify that an intermediate feature truly does not exist, rather than that we are simply unable to detect it. This issue lies at the very core of the paper’s argument, but remains insufficiently supported both logically and methodologically.

3. The overall argument remains at a high-level philosophical discussion, without formal logical analysis or empirical grounding. While the thought experiments are interesting and illuminating, they are not rigorous enough to substantiate the central claim of an ontological discontinuity between causation and correlation.

**Questions:**

Please see weakness.

---

### Note · Authors · 2025-12-21

I have read and agree with the venue's withdrawal policy on behalf of myself and my co-authors.